# Equity Analysis of Repeated Cross-Sectional Survey Data on Mental Health Outcomes in Saskatchewan, Canada during COVID-19 Pandemic

**DOI:** 10.3390/ijerph192113808

**Published:** 2022-10-24

**Authors:** Nazeem Muhajarine, Daniel A. Adeyinka, Vaidehi Pisolkar, Md Sabbir Ahmed, Natalie Kallio, Vithusha Coomaran, Tom McIntosh, Nuelle Novik, Bonnie Jeffery

**Affiliations:** 1Saskatchewan Population Health and Evaluation Research Unit (SPHERU), University of Saskatchewan, 104 Clinic Place, Saskatoon, SK S7N 2Z4, Canada; 2Department of Community Health and Epidemiology, College of Medicine, University of Saskatchewan, 107 Wiggins Rd, Saskatoon, SK S7N 5E5, Canada; 3Department of Politics and International Studies, University of Regina, 3737 Wascana Parkway, Regina, SK S4S 0A2, Canada; 4Faculty of Social Work, University of Regina, 3737 Wascana Parkway, Regina, SK S4S 0A2, Canada

**Keywords:** equity-seeking, COVID-19, anxiety, depression, mental health care, Saskatchewan

## Abstract

This paper aims to understand the impact of COVID-19 on three mental health outcomes—anxiety, depression, and mental health service use. Specifically, whether the associations between social and economic variables and these outcomes are exacerbated or buffered among equity-seeking groups in Saskatchewan. We analyzed secondary datasets of Saskatchewan adults from population-based national surveys conducted by Mental Health Research Canada (MHRC) on three occasions: cycle 2 (August 2020), cycle 5 (February 2021), and cycle 7 (June 2021). We examined temporal changes in the prevalence of anxiety, depression, and service utilization. Using the responses from 577 respondents in cycle 5 dataset (as it coincides with the peak of 2nd wave), we performed multinomial logistic regression. The policy implications of the findings were explored empirically through a World Café approach with 30 service providers, service users and policy makers in the province. The prevalence of anxiety and depression remained steady but high. Mental health services were not accessed by many who need it. Participants reporting moderate or severe anxiety were more likely to be 30–49 years old, women, and immigrants who earned less than $20,000 annually. Immigrants with either college or technical education presented with a lesser risk of severe anxiety. Factors associated with moderate or severe depression were younger age (<50 years), low household income, as well as immigrants with lower levels of education. Racialized groups had a lower risk of severe depression if they were under 30 years. Students and retirees also had a lower risk of severe depression. Canadian-born residents were more likely to require mental health supports but were not accessing them, compared to immigrants. Our analysis suggests mental health outcomes and service utilization remain a problem in Saskatchewan, especially among equity-seeking groups. This study should help drive mental health service redesign towards a client-centred, integrated, and equity-driven system in Saskatchewan.

## 1. Introduction

Initially labelled the ‘great equalizer,’ the coronavirus disease 2019 (COVID-19) pandemic has turned out to be anything but the ‘great divider’ [1,2]. Although public health researchers expected the pandemic would be socially graded, common-sense belief was that ‘anyone can get COVID’ and misunderstood differential vulnerability raised by health disparities [2,3,4]. COVID-19 has been shown to increase the pre-existing health and economic inequities as well as creating new inequities [3]. For example, previous studies have shown the predominance of COVID-19 morbidity and mortality rates, regardless of pandemic wave or the variants, among population groups who are racial or ethnic minorities [5,6]; women [7]; those employed in the service sector or underemployed [5,7,8]; and who live in precarious housing [7,9].

Mental health at the population level is particularly sensitive to large social disruptions, for example, the 9/11 attacks, December 2004 Indian Ocean tsunami, Hurricane Katrina [10,11,12]. Since the beginning of the pandemic, several notable studies have reported COVID-19-related mental health impacts; specifically noting increasing mental health concerns across Canada [13,14]. The impact of COVID-19 on mental health outcomes has been systematically different in certain groups, mainly related to their social, economic, and cultural characteristics [15,16,17]. Therefore, the need is urgent to give special attention to the subgroups that are at greater risk of poorer mental health outcomes.

“The Rainbow Model” of the main determinants of health by Dahlgren and Whitehead, therefore, offers a framework to conceptualize the determinants of health for the overall population [18,19]. It provides a context to our analysis taking into consideration that COVID-19 impacts on mental health are related to biological, lifestyle, cultural, socioeconomic, and environmental factors. The model describes that individual lifestyle factors are rooted in the social and community networks, within living and working conditions, and further embedded in the socioeconomic and cultural conditions [18,19]. In terms of COVID-19, individual, family and community factors may increase exposure and susceptibility to the virus and provide decreased opportunity to recover from the disease due to limitations experienced with socioeconomic, cultural, and environmental factors—which can lead to social inequities as seen throughout the pandemic.

For example, women and girls have been particularly affected during the COVID-19 pandemic due to an overload of caregiving responsibilities, susceptibility to gender-based violence, and financial difficulties [20,21]. Younger populations as well as those with pre-existing mental health conditions and disabilities are seen to have poorer mental health outcomes and greater difficulties with coping and reduced access to, or disruptions in, mental health services due to physical distancing [13,16,22,23,24,25]. Those with economic hardships due to loss of employment during the pandemic have an increased likelihood of having higher levels of distress, anxiety, and depression [8,26].

Indigenous and ethnocultural groups have also faced disproportionate impacts of the COVID-19 pandemic. Indigenous communities reportedly have worse mental health outcomes as compared to non-Indigenous people. Living in rural or remote regions make it difficult to access care and having pre-existing health conditions adds to this complexity [15,24]. In Canada, as compared to whites, visible minorities—namely South Asian, Chinese, Black, Filipino, and Arab—were seen to have adverse mental health outcomes [27]. COVID-19 related discrimination and stigmatization, and barriers to accessing mental health services, contribute to poorer mental health outcomes in racialized minorities [17,28].

To unpack, quantify and access how social and economic factors (namely age, gender, education, and income) intersects with ethnocultural attributes (namely immigrant status; ethnocultural minority status, including Indigenous status; and physical disabilities)—hereafter referred to as equity-seeking groups—we pursued two lines of analysis.

First, we conducted an analysis of population level data in Saskatchewan. We aimed to understand the impact of COVID-19 on three mental health outcomes: anxiety, depression, and mental health service use. We hypothesize that social and economic differences observed in anxiety and depression outcomes and in accessing mental health services are inequitable, and therefore unjust, for equity-seeking groups compared to mainstream groups (i.e., interaction effects). We described the self-reported prevalence of these mental health outcomes and service use over three distinct points during the pandemic to date.

Second, we used the data from the height of the third wave of the pandemic in Saskatchewan, in early February 2021 as the basis for a structured policy dialogue that, first, validated the quantitative data with the lived experience of participants in the community mental health system; and, second, began to articulate a policy agenda aimed at correcting the shortcomings exposed by the pandemic experience.

## 2. Materials and Methods

### 2.1. Data Sources

This study analyzed the data collected by Mental Health Research Canada (MHRC) from their population-based national surveys. Since April 2020, MHRC has been leading a study to gauge the effects of COVID-19 on mental health outcomes among Canadians. MHRC releases a new survey every 6–8 weeks, with updated questions reflecting the current landscape of COVID-19, added to a core set of consistently repeated questions. Due to the negative impacts of lockdown measures on participants’ recruitment, data were collected from a randomly selected sample of adult Canadians via an online portal managed by Pollara Strategic Insights [29], with a sample size equivalent to a probability sampling of ±1.5% margin of error. To generate a representative sample of the Saskatchewan population, the samples were weighted based on age, gender, and location of residence as per Canada Census 2016.

This study used cycle 2 (21–31 August 2020), 567 respondents, cycle 5 (1–8 February 2021), 577 respondents, and cycle 7 (7–13 June 2021), 590 respondents, specifically focusing on adults 18 years and over in Saskatchewan. The choice of measurement points included in the analysis was informed by COVID-19 epidemic curves (see Appendix A). Study cycle 2 coincided with the end of the first wave of COVID-19 in Saskatchewan (i.e., 7-day average of 11 active cases per 100,000), cycle 5 with near-peak of the second COVID-19 wave (205 active cases per 100,000), and cycle 7 with the downward slope of the third COVID-19 wave (69 active cases per 100,000). The different time points included in the analysis also reflect the seasonal effects on the COVID-19, a respiratory-born disease, and its influence on different pandemic waves and in turn on the mental health outcomes.

### 2.2. Outcome Variables

The outcomes of interest were: (1) severity of anxiety; (2) severity of depression; and (3) mental health support. The respondents rated their level of anxiety and depression since the COVID-19 outbreak on a Likert scale ranging from 0 to 10. Based on pre-defined criteria [29], the severity of mental health outcomes was categorized as mild (0–4 ratings), moderate (5–7 ratings) and severe (8–10 ratings). The mental health support was categorized based on perceived mental health needs and access to care (grouped: not needed and did not access support, needed support and received, needed support but no access).

### 2.3. Independent Variables and Covariates

The variables pertaining to attributes of the equity-seeking groups, namely, migrant status (grouped: born outside Canada, born in Canada), ethnicity (grouped: identify as an ethnocultural minority, yes or no), and physical disability/impairment (identify as having, yes or no), were used as moderating variables. Immigration status was included in this study to delineate the effects of migration and settlement on mental health outcomes, while ethnicity was included to assess the effects of ethnic and cultural identity (other than Indigenous) on the selected outcome variables.

The covariates were sociodemographic factors such as gender (woman, man), age (grouped: 18–29, 30–49, 50+ years), location of residence (first three digits of postal code, grouped: rural, mid-sized towns/cities, Regina, Saskatoon), household composition (grouped: live alone, live with others); parental status (grouped: not a parent, has children under 18 years, has children 18 years and older, has children in both age categories), and employment status (grouped: employed, retired/student, unemployed), highest level of education (elementary/high school, college/technical, university), household income (<$20,000, $20,000–$49,999, $50,000–$99,999, $100,000 or more).

### 2.4. Data Analysis

The background characteristics of the study participants and the prevalence of mental health outcomes and mental health support-seeking behavior were reported as weighted frequencies and percentages. Chi-square tests were performed to compare the prevalence of outcome variables across the background characteristics of the study participants. To identify the factors associated with anxiety, depression, and mental health support-seeking behavior, we used the study cycle 5 dataset, collected in early February 2021, at a near-peak of the COVID-19 second wave in Saskatchewan.

Multinominal logistic regression models were fitted, and relative risk ratios (RRR) and their respective 95% Confidence Intervals (CI) were reported. Three separate multinomial regression models were generated using anxiety, depression, and mental health support as dependent variables. We observed that due to the smaller sample size of the Indigenous respondents in the data, the models did not converge (quasi-complete separation), and we wanted to retain all respondents, so this group was merged with the ethnocultural minorities. The covariates that showed *p* < 0.25 in the bivariate analyses were included in the multivariable models. Data were analyzed using STATA^TM^ v14.2 (StataCorp, College Station, TX, USA) and sampling weights were applied. All statistical tests were considered significant at *p* < 0.05.

### 2.5. Implications Identified through a World Café Policy Dialogue

Following the findings of the first two cycles (August 2020, and February 2021), our research team convened a gathering of 30 individuals from the mental health sector to discuss the impacts of the COVID-19 pandemic on the mental health of Saskatchewan adults and chart a research-informed course forward for post-pandemic mental health services. Attendees included representatives from community mental health agencies, health organizations, front line service providers, service recipients, and their advocates, and individuals working in policy and program development for the Saskatchewan Health Authority and the Ministry of Health.

A World Café format was used as a strategy to generate dialogue and ideas. World Café is recognized as an effective change management tool in a range of settings with a diverse array of stakeholders [30] that emphasizes strengths-based learning and mutual creativity [31,32]. Unable to meet face-to-face due to pandemic restrictions, we modified the traditional World Café format and process to hold a virtual gathering while minimizing the risk of ‘zoom fatigue.’ Indeed, one of the advantages of a virtual event is it allowed representation from across the province from people who might otherwise have been unable to travel to participate [33].

Table discussions on Day 1 focused on results of the first two of the three rounds of surveys discussed in this paper with an emphasis on, first, the lived experience of both accessing and delivering mental health services during the pandemic and, second, on what these experiences said to participants about how mental health services can or should be (re)organized in the province post-pandemic. The shorter Day 2 discussion, held as a plenary, allowed participants to review, validate, and build on a summary of the Day 1 discussions.

## 3. Results

Table 1 shows the background characteristics of the study participants for each of the three cycles. Most respondents were 50 years and above, lived with other family members, were employed, and were born in Canada.

Self-reported anxiety and depression in each study cycle are presented in Figure 1. During study cycles in August 2020 and June 2021, one respondent in five had severe anxiety and one in seven had severe depression. However, in the second, Alpha-driven, pandemic wave (early February 2021), one respondent in 4 (26.1%) reported severe anxiety and nearly one in four reported severe depression. Figure 1C illustrates the prevalence of accessing mental health support since the beginning of the pandemic in Saskatchewan. In June 2021, about 12% of respondents reported needing—but not accessing—mental health support. In February 2021, however, the highest percentage of respondents reported they needed mental health support and received it (23.9%). Prevalence of all three mental health outcomes (i.e., anxiety, depression, and mental health support) based on the demographic and socioeconomic status and equity-seeking characteristics of the study participants are presented in Appendix A.

Factors associated with anxiety during study cycle 5 are presented in Table 2. Age, gender, income, and educational status were significantly associated with anxiety. Respondents 30–49 years of age were 2.6 times more likely to report moderate anxiety (95% CI: 1.41 to 5.05), and 3.4 times more likely to report severe anxiety (95% CI: 1.62 to 7.44), compared to the respondents who were aged 50 years and above. Women were more likely to report moderate anxiety [RRR: 1.73 (95% CI: 1.01 to 2.94)] and severe anxiety [RRR: 2.31 (95% CI: 1.21 to 4.39)], compared to men. The effect of low income on moderate anxiety was higher for those born outside of Canada versus in Canada. Among migrants, at each increment of income level measured, the probability of moderate anxiety decreased, in a linear fashion, whereas for those born in Canada this pattern was not observed. The effect of education on severe anxiety was distinctly different, i.e., opposite, for migrants versus Canadian-born respondents. Among migrants, those who had the lowest level of education (i.e., elementary/secondary education) reported the highest probability of anxiety, whereas for Canadian-born, those who had a college/technical education level reported the highest probability of anxiety (Figure 2). A full regression model analysis is presented in Appendix A.

As shown in Table 3, age, employment status, and income were significantly associated with depression during study cycle 5 (February 2021). Younger respondents, i.e., 16–29 and 30–49 years, compared to 50 years and above, were 3.6 and 3.7 times more likely to report moderate depression, respectively. A similar pattern, but higher risk ratios, were observed for younger respondents and severe depression: 4.3 and 6.7 times more likely to report high depression for those 16–29 and 30–49 years, respectively. Income level also showed significant associations with severe depression. Compared to those who reported the highest income level ($100,000 or greater annually), those reporting less than $20,000 were 7.09 times more likely to report severe depression, those reporting $20,000–$49,999 were 9.8 times more likely, and 3.31 times more for those in the $50,000 up to $100,000 income bracket. Among migrants, those who had elementary/secondary education reported the highest probability of moderate depression, whereas for Canadian-born respondents, those who had a college/technical education reported the highest probability of moderate depression. The association between age and ‘severe’ depression was moderated by self-identification as an ethnocultural minority. For those who indicated they were not a minority group (i.e., Whites), the highest probability of ‘severe’ depression was observed among the youngest age group, 16–29 years. Among those who identified as visible minority, the highest probability of ‘severe’ depression was for those 30 years and older (Figure 3). A full regression model analysis is presented in Appendix A.

Table 4 shows the factors associated with access to mental health support reported in February 2021. Age, household composition, income, and immigration status were significantly associated with mental health support-seeking behavior. The likelihood of respondents reporting that they needed mental health support and had received it showed an inverse association with age (i.e., the lower the age, the higher the likelihood). Those living alone were 2.2 times more likely to report needing and receiving mental health support, compared with those living with other family members (95% CI: 1.14 to 4.44). Those respondents born outside Canada had an 86% less likelihood of reporting they needed mental health support but had not accessed it, compared to those born in Canada [RRR: 0.14 (95% CI: 0.05 to 0.43)]. A full regression model analysis is presented in Appendix A.

The detailed results of the World Café Policy Dialogue are contained in a report [33] which identified four key themes that ran through the discussions. First, the provincial system was simply unprepared for the extent and duration of the pandemic and lacked capacity to easily adjust service delivery modes. This exposed gaps between community mental health services and the health care system that could not be easily filled. Second, existing inequalities in society relating to race, gender, socio-economic status, and disability were worsened by the pandemic and the manner in which those inequalities both overlap and reinforce each other became more apparent. Third, in the ‘pivot’ to new ways of working, interacting, and accessing public services, significant issues in getting timely, accurate information to people emerged. As modalities of service changed in response to pandemic restrictions, it was difficult for clients to find out where those services were and how best to access them. It also sometimes meant preferred services were replaced with what some felt were fewer desirable options. Fourth—one silver lining among the clouds—was a sense the pandemic had highlighted, for many who might have been previously unaware, the need for greater attention to the mental health system from the public and policymakers. The ‘shared trauma’ of the pandemic brought to light the importance of mental health to overall health and well-being.

## 4. Discussion

Our analysis highlights the social factors contributing to inequitable distribution of anxiety and depression, and accessibility and/or inaccessibility to mental health supports—5, 11, and 15 months into the COVID-19 pandemic in Saskatchewan. Estimating the impacts of social inequities on mental health outcomes and supports will inform planning and implementation of targeted interventions, particularly during the post-COVID-19 period.

In this study, we observed that, while the prevalence of anxiety and depression remained steady since August 2020, mental health supports are inaccessible to, or not accessed, by many who need them in Saskatchewan. From this study, younger people (30–49 years), women, individuals from households with annual earnings less than $20,000, and those who migrated to Canada were more likely to report moderate or severe forms of anxiety. However, immigrants with college or technical education presented with a lesser risk of severe anxiety. Also, respondents younger than 50 years of age who had lower household income, as well as immigrants with lower levels of education, presented with significantly higher risks of moderate to severe depression. Racialized groups (i.e., Indigenous and ethnocultural minorities), in contrast, had a lower risk of severe depression if they were under 30. Students and retirees also had lower risk of severe depression. Surprisingly, Canadian-born residents were more likely to require mental health supports but were not accessing them, compared to immigrants.

Since the onset of COVID-19 in Canada, governments at all levels have implemented countermeasures as part of the emergency response to halt rapid community transmission. As the pandemic progressed, news of variants of concern and feelings of isolation from lockdown and social distancing measures impacted mental health [34,35,36]. Although the prevalence of anxiety and depression has stabilized, our study found evidence that anxiety and depression among Saskatchewan residents are higher compared to the pre-pandemic era. Prior to COVID-19, the percentage of adults reporting moderate to severe forms of anxiety and depression were 39% and 32%, respectively. By June 2021 (cycle 7), 58.1% of Saskatchewan adults reported having moderate to severe forms of anxiety (36.9% moderate and 21.2% severe). Also, 4 respondents in 10 (42%) indicated moderate to severe depression, declining by 6% from 44.1% in cycle 2 and 10% in cycle 5. A possible explanation for the consistent rates is the resilience of people developing coping mechanisms to adapt to the adversity required due to COVID-19 [34]. Emerging evidence suggests COVID-19 might be here to stay, and people are learning to live with it. Of concern are reports of individuals turning to health-damaging coping mechanisms, such as increased substance use. For instance, cannabis use has increased by 30% in Saskatchewan [37]. People with substance use issues are also more likely to report severe anxiety and depression.

We report that mental health support is suboptimal, despite the high prevalence of anxiety and depression. Across the three data collection cycles, we observed mental health supports were not accessible to about 12% of people in Saskatchewan who needed them [reference blinded]. Not only has COVID-19 negatively impacted mental health; some government countermeasures—particularly social isolation, physical distancing, and lockdown measures—have worsened pre-existing inequities in accessing mental health supports and treatment [29]. To minimize these effects, stakeholders are recognizing alternative means of providing care to individuals with mental health issues. For example, Wellness Together Canada, a free and confidential online platform that provides resources, peer-support counselling and self-care is an example of such interventions to help improve mental health of Canadians [38]. However, inadequate public awareness of such initiatives remains a big problem [29].

Given evidence pointing to negative psychological impact on the equity-seeking groups due to social disruptions [39], more innovations and nimble, adaptive responses, are needed to address the structural barriers in order to make specialized mental health services readily available. In the 7th cycle of the survey, reasons given by Saskatchewan respondents for not assessing mental health services (in order of importance) were inability to pay for treatment (27%), lack of confidence in the health system (27%), lack of awareness of available services and where to get them (23%), fear of stigma (17%), limited access to treatment (16%), interference with job (6%), and language barriers (3%). These findings are consistent with the rest of Canada [40].

One surprising finding was that immigrants who reported having mental health issues requiring treatment and services were more likely to access them, compared to Canadian-born respondents. This could be due to the contributions of social support networks that facilitated the integration of migrants in Canada. The pivotal role of support networks cannot be underestimated [41,42]. According to a recent study, immigrants and Canadian-born residents showed a comparable sense of belonging to their local communities; also, mental health-associated stigma is a systemic problem that affects both population groups [43]. In a pre-pandemic study, Saunders et al., (2018) [44] reported outpatient mental health services should be restructured to meet the specific needs of immigrants (particularly refugees). Further research is therefore recommended to assess whether mental health care services somehow contribute to buffering adverse mental health outcomes in most vulnerable migrant groups.

The ongoing economic impact of COVID-19 is evident in mental health outcomes among immigrants with low socioeconomic status; those who earn less than $20,000 annually are more vulnerable to moderate anxiety, and those with less than college degrees more prone to moderate depression. The double jeopardy of the short- and long-term impacts of COVID-19 on health outcomes and economic downturns among immigrants and their children have been documented in other high-income countries [45]. More economic opportunities must be created to mitigate the vulnerability of low-paid immigrants (who are also likely to have less stable employment and lower seniority) [45].

Another important finding from this study is that older adults (50 years or more) were less likely to report depression than younger adults. This finding is similar to other reports [46,47]. In some respects, this is not surprising given that older adults are generally more resilient to mental health disorders than younger people [48]. On the other hand, severe depression is nuanced by the intersection of ethnocultural minority group and age. The higher prevalence of mental health disorders among Indigenous and ethnocultural minority populations is well documented [49,50]; but it is surprising to see racialized groups in this study are less likely to report severe depression if they are under 30 years. This finding raises further questions. Could the impact of COVID-19 among the young, racialized individuals somehow have been buffered by the provision of age-appropriate and culturally safe interventions led by members of their own communities? If so, this could be a key strategy to engage this equity-seeking group.

The link between gender and anxiety is well documented, with women having increased risk of anxiety compared to men [51,52]. In this study as well, the risks of moderate and severe anxiety were elevated among women as compared to men. It is obvious women’s mental health should be prioritized. Also, future studies are needed to better understand the pathway of influence of gender on anxiety disorders.

In terms of beginning a concrete policy agenda for Saskatchewan, World Café participants articulated a consensus on some key principles for moving forward in changing how we organize and deliver mental health services in the province, some of which touch on old themes.

First, there is a need to keep breaking down silos inside government (e.g., between departments) and between the government, health system, and communities. The pandemic further highlighted the lack of coordination between these elements in terms of program design and delivery. Second, we need to recognize there were some innovative partnerships that arose during the pandemic to overcome some of the barriers and challenges noted above, and these need to be sustained and nurtured (and replicated, where appropriate) post-pandemic. Third, the pandemic emphasized and underscored the need for ‘flexibility’ and ‘innovation’ to be key elements of any service redesign or reform to ensure the right service in the right place for the right people at the right time.

This can only work if we take seriously the need for better intersectoral cooperation and collaboration and if we shift service away from its current focus on the service provider/agency and towards the patient/client. This client-centred focus is what will allow the system to better deal with the intersectionality of the economic and social determinants of health highlighted in the pandemic’s unequal impacts.

Undoubtedly the pandemic has strengthened the argument about the need to rethink mental health services in the province. As more data become available and the picture of post-COVID Saskatchewan comes into even sharper focus, further conversations--like that described here—will be integral. Data and a better sense of the policy and political landscape for change will provide an opportunity to focus on both the successes and challenges that emerged from the pandemic and, thus, to sketch what a community-based, person-focused system of mental health services could look like in a post-COVID Saskatchewan.

A limitation of this research is the potential for information bias due to the secondary nature of the dataset and self-reporting of mental health outcomes. However, it should be noted data were collected using previously validated questionnaires, by Mental Health Research Canada. Although, we attempted to ensure data representativeness by weighing the demographic and regional quotas using Census data, the findings from multinomial regression analysis may not be generalizable specifically to the Indigenous populations because they were not included in cycles 4 or 5 data collection. However, by comparing the distribution of the study sample with the Census population, we observed consistencies. For example, 51% of survey respondents were women compared with 50.3% of Saskatchewan adults. Also, the 11.3% of ethnocultural minority population in Saskatchewan (not inclusive of Indigenous populations) is consistent with the 10.5% reported for the Saskatchewan province.

It is imperative that future data collection efforts continue to include data on equity-deserving populations, that the data analyzed through equity-lens and where appropriate and possible data be shared with entities that champions equity and fairness issues.

## 5. Conclusions

This study provides the temporal trend of mental health outcomes, service utilization and their associated characteristics amid COVID-19 pandemic and social disruptions in Saskatchewan. We found compelling evidence that the percentages of Saskatchewan residents reporting moderate to severe forms of anxiety and depression are stable, but still high. Also, there are residents who experienced anxiety and/or depression and could not access care. Based on the study findings, stakeholders need to target the marginalized populations identified in this paper with age-, gender- and culturally responsive interventions. Also, continuous economic interventions during this pandemic will reduce the negative impacts of mental health on equity-seeking groups.

With reference to the policy dialogue conducted, the preliminary nature of the policy agenda that emerged from the discussions, taking place as it did at the height of the third wave, should be recognized. This speaks to the need to continue the engagement and dialogue in subsequent similarly structured events. As subsequent data makes the lasting mental health impacts more apparent, so too will subsequent dialogues give more concrete shape to the needed policy changes and provide direction on how best they can be achieved.

## Figures and Tables

**Figure 1 ijerph-19-13808-f001:**
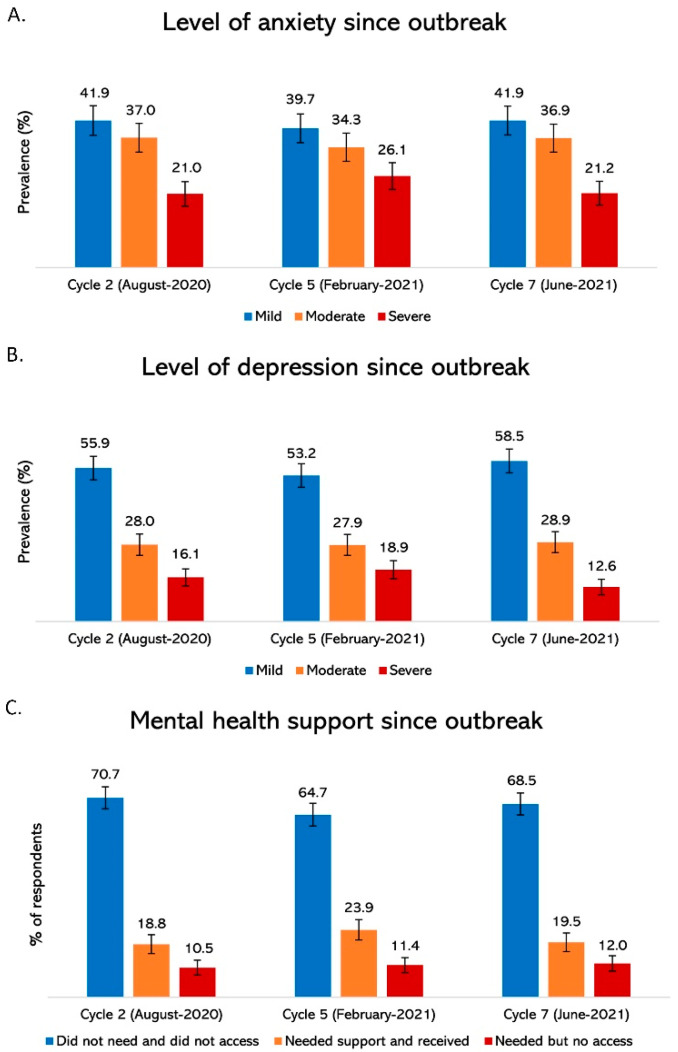
Prevalence of mental health support and access to care in Saskatchewan: (**A**) anxiety; (**B**) depression; and (**C**) mental health support.

**Figure 2 ijerph-19-13808-f002:**
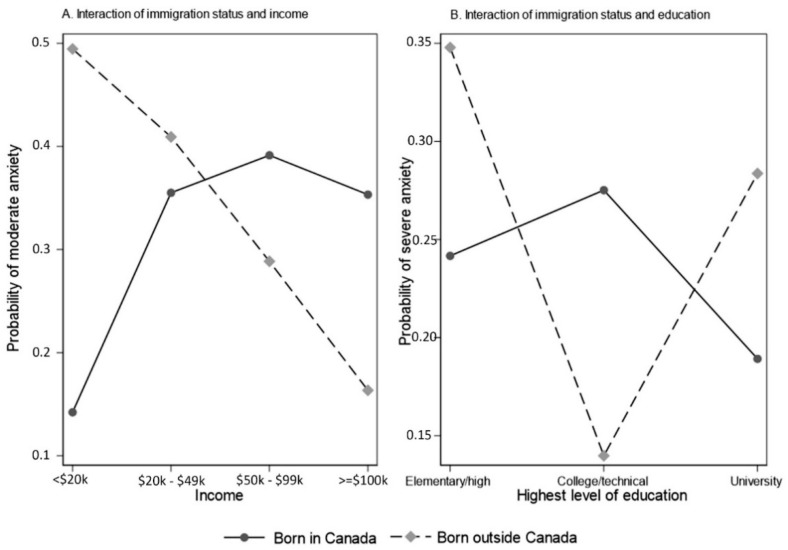
Interaction plots for anxiety: (**A**) moderate anxiety, income effects moderated by immigration status; and (**B**) severe anxiety, education effects moderated by immigration status.

**Figure 3 ijerph-19-13808-f003:**
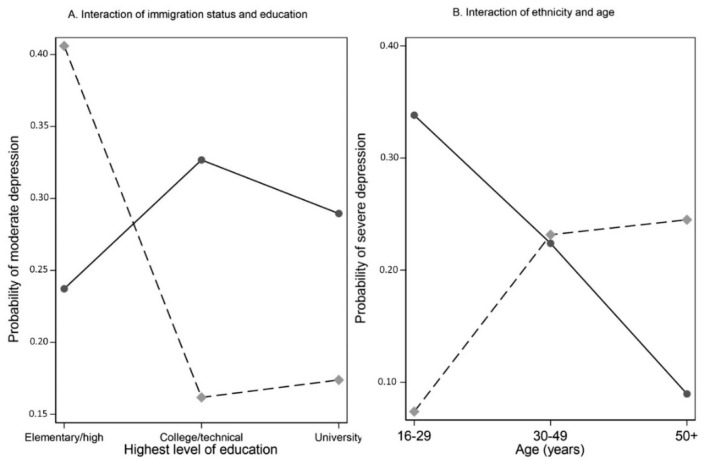
Interaction plots for depression: (**A**) moderate depression, education effects moderated by immigration status; and (**B**) severe depression, age effects moderated by ethnicity.

**Table 1 ijerph-19-13808-t001:** Characteristics of the Mental Health Research Canada—Saskatchewan study samples, August 2020, February 2021, and June 2021.

Variables	Weighted Frequency ^1^ (%)		
	Study Cycle 2, August 2020 (N = 576)	Study Cycle 5, February 2021 (N = 577)	Study Cycle 7, June 2021 (N = 590)
Current age (years)			
16 to 29	103 (17.9)	104 (18.0)	118 (20.0)
30 to 49	202 (35.1)	212 (36.8)	217 (36.8)
50 and above	271 (47.0)	261 (45.3)	255 (43.2)
Gender			
Women	292 (50.8)	293 (50.8)	299 (50.7)
Men	281 (48.8)	277 (48.0)	288 (48.8)
Not disclosed	2 (0.4)	7 (1.3)	3 (0.5)
Location of residence			
Mid-size city/town	80 (13.9)	90 (15.6)	77 (13.0)
Rural	144 (25.0)	145 (25.1)	174 (29.5)
Regina	189 (32.9)	167 (28.9)	175 (29.7)
Saskatoon	162 (28.2)	175 (30.4)	164 (27.7)
Household composition			
Live alone	136 (23.6)	121 (21.0)	122 (20.7)
Live with others	440 (76.4)	456 (79.0)	468 (79.3)
Parental status			
Not a parent	354 (61.4)	357 (61.8)	346 (58.7)
Has children ≤ 17 years	133 (23.1)	134 (23.3)	143 (24.3)
Has children ≥ 18 years	71 (12.3)	72 (12.5)	88 (14.9)
Has children in both age groups	18 (3.2)	14 (2.4)	12 (2.1)
Employment status			
Unemployed	65 (11.3)	62 (10.7)	51 (8.6)
Retired/Student	153 (26.5)	165 (28.7)	171 (29.0)
Employed	341 (59.2)	347 (60.1)	358 (60.8)
Not disclosed	17 (2.9)	3 (0.5)	9 (1.6)
Highest level of education			
Elementary/high	169 (29.3)	166 (28.8)	182 (30.9)
College/technical	187 (32.5)	179 (31.1)	171 (28.9)
University	209 (36.3)	225 (38.9)	227 (38.6)
Not disclosed	11 (1.9)	7 (1.1)	10 (1.6)
Income			
<$20K	38 (6.6)	40 (6.9)	22 (3.7)
$20K to $49K	121 (21.0)	142 (24.6)	116 (19.6)
$50K to $99K	225 (39.0)	190 (32.9)	250 (42.4)
≥$100K	135 (23.4)	160 (27.8)	155 (26.3)
Not disclosed	58 (10.0)	45 (7.8)	47 (8.0)
Immigration status			
Born outside Canada ^2^	145 (25.1)	137 (23.7)	136 (23.0)
Born in Canada ^3^	419 (72.8)	417 (72.3)	443 (75.1)
Not disclosed	12 (2.1)	23 (4.0)	
Ethnicity			
Indigenous/Ethnocultural minority ^4^ (African, Asian, Hispanic/Latino/Others)	Not available	65 (11.3)	101 (17.2)
White		512 (88.7)	489 (82.8)
Physical disability			
Yes	Not available	33 (5.7)	37 (6.3)
No		544 (94.3)	553 (93.7)

^1^ Weighted number are reported after rounding; ^2^ Born outside Canada include all respondents who were born outside Canada irrespective of ethnocultural identity; ^3^ Born in Canada include all respondents born in Canada; ^4^ Data for Indigenous was available only in cycle 7.

**Table 2 ijerph-19-13808-t002:** Factors associated with self-reported moderate and severe anxiety compared to mild anxiety, cycle 5 MHRC survey (1–8 February 2021), Saskatchewan.

	Moderate Anxiety		Severe Anxiety	
	RRR ^1^ (95% CI)	*p*-value	RRR ^1^ (95% CI)	*p*-value
Age (years)				
16–29	1.47 (0.60–3.57)	0.388	1.97 (0.69–5.61)	0.200
30–49	2.67 (1.41–5.05)	0.002	3.47 (1.62–7.44)	0.001
50 and above	Reference		reference	
Gender				
Women	1.73 (1.01–2.94)	0.043	2.31 (1.21–4.39)	0.010
Men	Reference		reference	
Income				
<$20K	0.40 (0.11–1.46)	0.168	2.44 (0.78–7.63)	0.122
$20K to $49K	1.73 (0.83–3.62)	0.142	3.81 (1.52–9.50)	0.004
$50K to $99K	1.26 (0.68–2.35)	0.451	1.24 (0.53–2.85)	0.611
≥$100K	Reference		reference	
Educational status				
Elementary/high	0.47 (0.24–0.93)	0.032	1.01 (0.45–2.25)	0.977
College/technical	1.10 (0.61–1.99)	0.729	1.83 (0.85–3.93)	0.116
University	Reference		reference	
Income × immigration status				
<$20K × born outside Canada	29.24 (1.42–601.41)	0.029	2.34 (0.09–56.48)	0.600
$20K to $49K × born outside Canada	3.17 (0.55–18.03)	0.192	0.73 (0.10–5.04)	0.753
$50K to $99K × born outside Canada	1.78 (0.37–8.45)	0.467	0.97 (0.18–5.16)	0.979
≥$100K × born in Canada	Reference		reference	
Education × immigration status				
Elementary/high × born outside Canada	2.06 (0.47–9.04)	0.337	1.36 (0.29–6.21)	0.687
College/technical × born outside Canada	0.78 (0.22–2.81)	0.712	0.19 (0.03–0.95)	0.044
University x born in Canada	Reference		reference	

^1^ Relative Risk Ratio. Parsimonious model shown; adjusted for age, gender, employment status, income, educational status, physical disability, immigration status, and interactions: Age × immigration status, Gender × immigration status, Employment × immigration status, Income × immigration status, Education × immigration status.

**Table 3 ijerph-19-13808-t003:** Factors associated with self-reported moderate, and severe depression compared to mild depression, cycle 5 MHRC survey (1–8 February 2021), Saskatchewan.

	Moderate Depression		Severe Depression	
	RRR ^1^ (95% CI)	*p*-value	RRR ^1^ (95% CI)	*p*-value
Age (years)				
16–29	3.69 (1.48–9.20)	0.005	4.38 (1.31–14.66)	0.016
30–49	3.79 (1.78–8.09)	0.001	6.79 (2.72–16.89)	<0.001
50 and above	Reference		reference	
Employment status				
Unemployed	0.76 (0.34–1.70)	0.516	1.07 (0.46–2.50)	0.860
Retired/Student	0.67 (0.34–1.31)	0.247	0.38 (0.15–0.96)	0.041
Employed	Reference		reference	
Income				
<$20K	1.51 (0.40–5.60)	0.538	7.09 (1.75–28.72)	0.006
$20K to $49K	3.20 (1.47–6.97)	0.003	9.82 (3.24–29.71)	<0.001
$50K to $99K	1.68 (0.87–3.24)	0.121	3.31 (1.28–8.54)	0.013
≥$100K	Reference		reference	
Education × immigration status				
Elementary/high × born outside Canada	8.38 (1.60–43.91)	0.012	4.04 (0.63–25.72)	0.138
College/technical × born outside Canada	0.69 (0.15–3.16)	0.637	0.78 (0.12–4.86)	0.795
University × born in Canada	Reference		reference	
Age × ethnocultural				
16–29 × indigenous/ethnocultural minority	0.23 (0.00–7.63)	0.414	0.00 (0.00–0.38)	0.015
30–49 × indigenous/ethnocultural minority	1.88 (0.25–13.65)	0.532	0.35 (0.06–1.98)	0.238
50 and above × White	Reference		reference	

^1^ Relative Risk Ratio. Parsimonious model shows; adjusted for age, household composition, parental status, employment status, income, educational status, physical disability, immigration status, ethnocultural minority, and interactions: Age × Immigration status, Income × immigration status, Education × immigration status, Age × ethnocultural minority.

**Table 4 ijerph-19-13808-t004:** Factors associated with self-reported mental health support seeking behavior (needed support and received, needed but no access) compared to did not need and did not access, cycle 5 MHRC survey (1–8 February 2021), Saskatchewan.

	Needed Support and Received		Needed but No Access	
	RRR ^1^ (95% CI)	*p*-value	RRR^1^ (95% CI)	*p*-value
Age (years)				
16–29	6.02 (2.40–15.13)	<0.001	2.07 (0.80–5.32)	0.129
30–49	4.21 (2.05–8.67)	<0.001	1.35 (0.61–2.99)	0.499
50 and above	Reference		reference	
Household composition				
Live alone	2.25 (1.14–4.44)	0.018	0.85 (0.34–2.12)	0.741
Live with others	Reference		reference	
Income				
<$20K	3.57 (1.22–10.43)	0.020	1.05 (0.23–4.69)	0.947
$20K to $49K	2.75 (1.19–6.34)	0.018	2.48 (1.07–5.74)	0.034
$50K to $99K	1.45 (0.71–2.96)	0.304	1.15 (0.53–2.47)	0.719
≥$100K	Reference		reference	
Immigration status				
Born outside Canada	0.61 (0.32–1.14)	0.127	0.14 (0.05–0.43)	0.001
Born in Canada	Reference		reference	

^1^ Relative Risk ratio. Parsimonious model shown; adjusted for age, gender, household composition, employment status, income, educational status, physical disability, immigration status, ethnocultural minority.

## Data Availability

The dataset used for this study was collected, created, and is owned by Mental Health Research Canada (MHRC), and shared with the Authors under the MHRC Polling Data Sharing Agreement for COVID-19 Mental Health Polling Project. As required by the University of Saskatchewan Data Management Policy, the dataset for this study is available from N.M. or the USask’s ethics (ethics.office@usask.ca) on written request.

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
