# Peer review of "Equity Analysis of Repeated Cross-Sectional Survey Data on Mental Health Outcomes in Saskatchewan, Canada during COVID-19 Pandemic"

_ijerph, 2022, doi:10.3390/ijerph192113808_

Round 1

Reviewer 1 Report

The article entitled: COVID-19 and Mental Health Outcomes--Great equalizer it is not: Evidence from an equity analysis of repeated cross-sectional survey data from Saskatchewan, Canada delivers relevant information and scientific evidence about health equity and mental disorders during COVID-19 pandemic.

The overall manuscript, proposal and methodologies are carefully presented and very well explained and provide valuable information to followup decisions regarding mental health service delivery.

Some brief suggestions for authors:

1. Materials and Methods: the results section presents a table (1) with sample size per cycle, it is not clear in -data sources- section if the sum of all samples represent a N or sample size is a different number gathered from the general dataset.

2. Discussion can be strengthened with results from other studies, for example those used in the introduction or by studying others:
Torres González, C., Galindo-Aldana, G., García León, I. A., Alfredo Padilla-López, L., Alvarez Núñez, D. N., & Espinoza Gutiérrez, Y. I. (2020). COVID-19 voluntary social isolation and its effects in sociofamily and children's behavior. Salud Mental, 43(6), 263-271. https://libcon.rec.uabc.mx:4440/10.17711/SM.0185-3325.2020.036.

Author Response

Reviewer #1

R.1.1: The article entitled: COVID-19 and Mental Health Outcomes--Great equalizer it is not: Evidence from an equity analysis of repeated cross-sectional survey data from Saskatchewan, Canada delivers relevant information and scientific evidence about health equity and mental disorders during COVID-19 pandemic. The overall manuscript, proposal and methodologies are carefully presented and very well explained and provide valuable information to followup decisions regarding mental health service delivery.

Response: Thank you for the comments.

Changes made to the manuscript: none

R.1.2: Materials and Methods: the results section presents a table (1) with sample size per cycle, it is not clear in -data sources- section if the sum of all samples represent a N or sample size is a different number gathered from the general dataset.

Response: The range of the sample size for the three data cycles were provided in the methods—it is a repeated cross-sectional design. As recommended, we have provided sample size used for each data collection cycle.

Changes made to the manuscript: Page 4, lines 136-138: “The sample size for cycle 2 is 567 respondents, 577 respondents for cycle 5, and 590 respondents for cycle 7. In each cycle, adult Saskatchewan adults 18 years and over were recruited.”

R.1.3. Discussion can be strengthened with results from other studies, for example those used in the introduction or by studying others:

Torres González, C., Galindo-Aldana, G., García León, I. A., Alfredo Padilla-López, L., Alvarez Núñez, D. N., & Espinoza Gutiérrez, Y. I. (2020). COVID-19 voluntary social isolation and its effects in sociofamily and children's behavior. Salud Mental, 43(6), 263-271. https://libcon.rec.uabc.mx:4440/10.17711/SM.0185-3325.2020.036

Response: The suggested paper has been included in the discussion.

Changes made to the manuscript: Page 12, lines 352-354: “As the pandemic progressed, news of variants of concern and feelings of isolation from lockdown and social distancing measures impacted mental health (Torres et al. 2020; Peterson 2020; Samji et al. 2021).”

Page 12, lines 361-363: “A possible explanation for the consistent rates is the resilience of people developing coping mechanisms to adapt to the adversity required due to COVID-19 (Torres et al. 2020)”

Reviewer 2 Report

The design of the thesis is correct, the applied research methods do not raise objections. The problem and research goal are formulated clearly and precisely. From a technical point of view, the article is correct and pleasant to read.

Weaknesses:

- the originality of the presented research results is average

- the research methods / tools used are predictable

Strengths:

- recommendations are described in detail

- the ability to perceive and interpret social, cultural, economic and political relations

- interesting and extensive discussion of research results, focus on the need for political dialogue

To sum up: a good study that addresses mental health issues. In the future, it is worth working on greater originality of the research methodology.

Author Response

Reviewer #2

R.2.1: The design of the thesis is correct, the applied research methods do not raise objections. The problem and research goal are formulated clearly and precisely. From a technical point of view, the article is correct and pleasant to read.

Response: Thank you for the comments

Changes made to the manuscript: none.